# The Ratio of cf-mtDNA vs. cf-nDNA in the Follicular Fluid of Women Undergoing IVF Is Positively Correlated with Age

**DOI:** 10.3390/genes14071504

**Published:** 2023-07-23

**Authors:** Georgia Tsirka, Athanasios Zikopoulos, Kyriaki Papageorgiou, Charilaos Kostoulas, Ioannis Tsigkas, Efthalia Moustakli, Aris Kaltsas, Eleftheria Sarafi, Theologos M. Michaelidis, Ioannis Georgiou

**Affiliations:** 1Laboratory of Medical Genetics in Clinical Practice, Faculty of Medicine, School of Health Sciences, University of Ioannina, 45110 Ioannina, Greece; g.tsirka@uoi.gr (G.T.); chkost@uoi.gr (C.K.); ef.moustakli@uoi.gr (E.M.); 2Obstetrics and Gynecology, Royal Cornwall Hospital, Truro TR1 3LQ, UK; thanzik92@gmail.com; 3Department of Biological Applications & Technology, School of Health Sciences, University of Ioannina, 45110 Ioannina, Greece; kyriaki.papageorgiou@bri.forth.gr (K.P.); e.sarafi@uoi.gr (E.S.); tmichael@uoi.gr (T.M.M.); ioannis.tsigkas@bri.forth.gr (I.T.); 4Biomedical Research Institute, Foundation for Research and Technology—Hellas, 45115 Ioannina, Greece; 5Laboratory of Spermatology, Department of Urology, Faculty of Medicine, School of Health Sciences, University of Ioannina, 45110 Ioannina, Greece; a.kaltsas@uoi.gr

**Keywords:** follicular fluid (FF), mitochondria, cf-mtDNA/cf-nDNA, age, cf-mtDNA, cf-nDNA, IVF, ART, PCOS

## Abstract

Age-related mitochondrial markers may facilitate the prognosis of artificial reproductive technology outcomes. In this report, we present our study concerning the ratio of cf-mtDNA/cf-nDNA, namely the amount of cell-free mitochondrial DNA relative to cell-free nuclear DNA, in the follicular fluid (FF) of women undergoing IVF, aiming to generate a molecular fingerprint of oocyte quality. The values of this ratio were measured and compared among three groups of women (101 in total): (A) 31 women with polycystic ovary syndrome (PCOS), (B) 34 women younger than 36 years, and (C) 36 women older than 35 years of age. Real-time quantitative PCR (qPCR) was performed to quantify the ratio by using nuclear- and mitochondrial-specific primers and analyzed for potential correlation with age and pregnancy rate. Our analysis showed that the level of FF-cf-mtDNA was lower in the group of advanced-age women than in the groups of PCOS and non-PCOS women. Moreover, a significant positive correlation between FF-cf-mtDNA and the number of mature (MII) oocytes was observed. Collectively, the data show that the relative ratio of cf- mtDNA to cf-nDNA content in human FF can be an effective predictor for assessing the corresponding oocyte’s age-related performance in IVF.

## 1. Introduction

### 1.1. Cell-Free DNA

Mandel and Metais first identified cell-free DNAs (cfDNAs) in human serum in 1948 [1,2]. DNA fragments known as cell-free DNAs (cfDNAs) are dispersed into extracellular surroundings. They are defined as DNA fragments that exploded via apoptosis, necrosis, and active releasing mechanisms [3,4] fueled both from the nucleus and mitochondria [5]. Their length is thought to range from 40 to 200 base pairs (bp), peaking at 166 bp [6], about the size of the DNA that is wrapped around a nucleosome [7]. However, larger segments of up to >30 kb have also been found [8]. In general, longer DNA fragments > 10 kb are thought to be the result of necrotic cell death, such as those of cells in necrotized tumour tissue. The level of cfDNA in the blood varies greatly; it ranges from 0 to 100 ng/mL in healthy individuals and from 0 to 5 to >1000 ng/mL in cancer patients [6].

This type of DNA has been discovered in a variety of bodily fluids. cfDNA fragments can result from both healthy and diseased cells and are able to reveal vital details about a person’s health. Blood, urine, saliva, spinal fluid, semen, and follicular fluid have all been found to contain cfDNA, which is also in abundance in the physiological extracellular environment [8,9,10].

Mitochondria are found in the cytoplasm of eukaryotic cells and are responsible for energy production. They multiply independently of cell division and contain their unique genome, which is double-stranded circular DNA, the mitochondrial DNA (or mtDNA). mtDNA differs from nuclear DNA in several ways, such as the length, the chemical structure, and the number of copies. Unlike nuclear DNA, which is inherited from both parents, mitochondrial DNA is only maternally inherited but rare exceptions may occur [11,12,13].

mtDNA contains a heavy (H) strand (purine-rich) and a light (L) strand (pyrimidine-rich). In human cells, the mt-DNA is a 16,569 base-pair long molecule that encodes the 22 tRNAs, 2 rRNAs, and 13 polypeptides which are necessary for the synthesis of adenosine triphosphate (ATP) [14].

Mammalian cells can contain from a few hundred to thousands of mitochondria, depending on their size and energy requirements. One of the cell types harbouring the largest concentrations of mitochondria and mtDNA is the mature human oocyte, with approximately 100,000–600,000, while most other cells in the human body have about 100–10,000 mitochondria [15]. During fetal development, oocyte mitochondria begin to replicate; each oogonia cell contains about 200 mitochondria. An oocyte stalled at metaphase II contains roughly 100,000 mitochondria and between 50,000 and 550,000 copies of the DNA immediately before fertilization because replication proceeds in synchronization with maturation [11,16,17]. This is possibly due to the large energy demands of fertilization, syngamy, cleavage, and early embryonic development [18].

### 1.2. Follicular Fluid

The cf-mtDNA has been found in a variety of bodily fluids [19,20], including follicular fluid. Follicular fluid offers a rich research biological material because it represents the microenvironment required for the maturation of the oocytes. As a result, some of its biochemical properties may be crucial to the quality of the oocyte as well as to its likelihood of successful fertilization and fetal development. Oocyte quality is a significant determinant of embryo developmental ability and clinical pregnancy percentage, although, in most IVF facilities, the evaluation of oocyte quality is primarily restricted to an assessment of the morphological parameters. Promising biomarkers have been found in granulosa cells and follicular fluid according to several studies [21,22]. CfDNA concentrations in human follicular fluid were found to be substantially correlated with the viability of the oocytes and embryos and may be a potential, noninvasive diagnostic tool to enhance the success of in vitro fertilization (IVF).

According to a previous research report, granulosa cells (GC) actively released cf-mtDNA into the medium in response to mitochondrial malfunction. This relationship between cf-nDNA and cf-mtDNA and the cfDNA concentration in FF was also demonstrated [23]. Furthermore, it is still uncertain if the quantity of cf-n- and/or cf-mtDNA reflects the oocytes’ capacity for development. For instance, it has been found that low cf-mtDNA concentration in FF, but not low cf-nDNA, is associated with good oocyte development potential [24].

Follicular atresia has historically been associated with GC apoptosis. Nevertheless, recent research has shown that follicular atresia can activate various types of programmed cell death (PCD), primarily in GCs, including autophagy. Although different mitophagy signalling pathways have different roles depending on the types of cells or outside factors. Several types of autophagy result in the breakdown of particular organelles, such as mitophagy, for the destruction of mitochondria. Mitophagy, a mechanism by which cells selectively eliminate extra or damaged mitochondria by autophagy, is essential for maintaining mitochondrial homeostasis and cellular survival [25].

Aberrant mitochondria can be eliminated by mitophagy in the early stages of oogenesis. Despite the existence of mitophagy regulators in oocytes, after oocyte formation, mitophagy is not completely activated to clear defective mitochondria. As a result, malfunctioning mitochondria are transferred from the oocyte to the embryo. However, by increasing mitophagy, GC in the vicinity of oocytes can enhance mitochondrial activity, thus improving oocyte developmental potential [26].

### 1.3. The Requirement for ART Procedure Indicators

In order to figure out the cfDNA potential for the intended uses, researchers have recently investigated the cfDNA profile within its primary sources in reproductive medicine, such as semen, follicular fluid (FF), serum, and blastocyst [27,28]. Follicular fluid and granulosa cells constitute the unique environment for follicular maturation, oocyte development, and embryo development capacity and, thus, represent reliable oocyte and embryo biomarkers that could be used as supplementary prognostic/diagnostic tools useful for assisted reproduction techniques [29].

Several studies revealed that the amount of cfDNA in human follicular fluid correlated with embryo quality and could be used as a new biomarker, mainly in an assisted reproduction laboratory. More specifically, the amount of cf-nDNA detected in the follicular fluid has been shown to reflect granulosa cell apoptosis, while the cf-mtDNA content reflects the change in mitochondrial function and granulosa cell dynamics [30,31]. Based on several studies, women who underwent unsuccessful IVF treatments had significantly higher levels of cf-mtDNA in their follicular fluid than women who were successful in getting a positive pregnancy test. Women with poor oocyte quality also had significantly higher levels of cf-mtDNA in their follicular fluid than women with good oocyte quality [9,32].

The present study is a contribution to the existing research on markers in IVF techniques. Consequently, we focused on the follicular fluid cf-mtDNA/cf-nDNA ratio levels, also known as the amount of mitochondrial DNA about nuclear DNA, and their role in oocyte quality, as well as its potential involvement in the successful outcome of pregnancy resulting from IVF.

## 2. Materials and Methods

### 2.1. Patients’ Characteristics and Follicular Fluid Collection

A total of 101 women undergoing IVF/ICSI were included in our study. The samples of the follicular fluid were obtained from the IVF Unit in the Department of Obstetrics and Gynecology at the University Hospital of Ioannina. Only follicular-fluid samples free of blood were used in the investigation. There were 31 PCOS patients and 70 without any certain reproductive disorder. The latter were divided into two categories according to their age: women of the first reproductive age up to 35 years age and women of the second reproductive age, older than 35 years of age. The informed consent was signed by each participant. Rotterdam criteria were used to identify PCOS disease [33]. The Rotterdam PCOS diagnostic criteria in women include two of the following three characteristics: menstrual irregularity, hyperandrogenism, and polycystic ovarian morphology on ultrasonography [34]. The control group consisted of women aged up to 35 years old who had successful pregnancies. The patient’s age was between 23–43. Other characteristics information was collected about each participant: age, egg count, number of cumulus cells, and pathologies such as polycystic ovary syndrome (PCOS) [Table 1]. Several hormone-level estimations were performed such as follicle-stimulating hormone [FSH], luteinizing hormone [LH], and estradiol [E2] at day 2 or 3 of the cycle. Anti-Mullerian hormone [AMH] was also measured in all women. All hormones were estimated using the ELISA technique [Table 2]. All 101 patients had received antagonist treatment and ovarian stimulation with recombinant human FSH (brand name: Gonal F) or human menopausal gonadotropin (hMG) (brand name: Menopur). Ovarian stimulation response was monitored by measuring 17-β estradiol [E2] concentration.

### 2.2. cfDNA Extraction and Quantification

Follicular fluid was centrifuged at 5000 rpm for 5 min to get rid of granulosa cells, blood cells, and any other small cells contained in the fluid and the supernatant was collected. To maximize the performance of cfDNA-based assays, prior to DNA extraction, each pool of FF was centrifuged at 10,000× *g* for 15 min and the supernatant was transferred to a new Eppendorf tube. A volume of 200 μL from each FF Eppendorf was processed for cfDNA extraction by PureLink Genomic DNA mini kit (Invitrogen, Waltham, MA, United States) according to the manufacturer’s instructions. The extracted cfDNAs were eluted in 50 μL of elution buffer and quantified by NanoDrop Spectrophotometer (Thermo Fisher Scientific, Waltham, MA, USA) for measuring purity and concentrations. The purity of the extracted DNA was assessed by my measuring the A260/A80 ratio via spectroscopy for cfDNA analysis, both nuclear and mitochondrial-specific primers were used. For cf-nDNA quantification, we used the RPE primers 5′-ATAGGAAGCCAGAGAAGAGAGACT-3′ and 5′-TCTATCTCTGCGGACTTTGAGCAT-3′, (200 bp), corresponding to the human retinoid isomerohydrolase gene (RPE65), as a nuclear DNA reference gene and, for the cf-mtDNA quantification, we used the pair of primers 5′-TAGAGGAGCCTGTTCTGTAATCG-3′ and 5′-TAAGGGCTATCGTAGTTTTCTGG-3′, (205 bp), corresponding to a portion of the mitochondrially encoded 12 S RNA, as the mitochondrial DNA reference gene, as described in [Table 3].

### 2.3. Real-Time Quantitative-PCR (qPCR)

The cf-mtDNA/cf-nDNA ratio was evaluated by using quantitative polymerase chain reaction (qPCR). In summary, we mixed 0.02–0.03 μg of extracted cf-DNA, 0.25 μL of 10 μM of each primer, forward and reverse, and 5 μL of SYBR Green Master Mix (PowerUp SYBR Green Master Mix, Applied Biosystems, Thermo Fisher Scientific, USA), and 2.5 μL of distilled water, in a total volume of 10 μL for each reaction. The reaction was run on a Corbett Rotor-Gene 3000 Real-Time Rotary Analyzer (Corbett Research, Sydney, Australia) with the following cycling conditions: 50 °C for 2 min (melt 1), 95 °C for 2 min (melt 2), followed by 40 cycles of denaturing at 95 °C for 15 s and 60 °C for 1 min. Each sample was analyzed in duplicate. Subsequently, the ratio of cf-mtDNA to cf-nDNA was calculated by the ΔΔCt method [35,36], as described below, and used as a measure of the cell-free mtDNA content in each FF sample.

For the relative quantification (also known as the ΔΔCt method), we assigned a control group and used the cycle threshold Ct values as follows:(a)ΔCt for every sample as described above, by subtracting the average nuclear Ct value from the average mitochondrial Ct value: ∆Ct = (mtDNA Ct − nDNA Ct);(b)the mean ΔCt value for the control group, in our case, the control group consisted of reproductively younger women (≤35 years of age) who had a positive pregnancy outcome and without any reproductive disorders;(c)the ΔΔCt for each sample by subtracting the ΔCt of the control group from the mean ΔCt of the sample, that is ΔΔCt = ΔCt of a sample of interest − ΔCt of the control group [36];(d)the fold difference using the formula 2^−ΔΔCt^.

The ΔΔCt values for each sample by subtracting the ΔCt of the control group from the mean ΔCt of the sample, that is ΔΔCt = ΔCt of a sample of interest minus the ΔCt of the control group. As a control group, we considered the group of reproductively younger women (≤35 years of age) who had a positive pregnancy outcome and no reproductive disorders. The corresponding values (average ± SD) were as follows: for women ≤ 35, nonpregnant = 5065.4 ± 6414.7 and pregnant = 3904.6 ± 2863.4; for women > 35, nonpregnant = 2159.8 ± 6045.3 and pregnant = 540.2 ± 440.7; for PCOS, nonpregnant = 4084.2 ± 4334.9 and pregnant: 13344.3 ± 19621.4.

### 2.4. Statistical Analysis

For analyzing the data between two groups, we used Student’s *t*-test, and the data from three or more groups were analyzed using analysis of variance (ANOVA). Export data, the mt-DNA copy numbers, and ΔCt rates were placed on Microsoft Excel spreadsheets for graph design. *p* < 0.05 was considered statistically significant.

## 3. Results

The age groups of women undergoing IVF and women with PCOS cf-mtDNA in follicular fluid had a positive correlation with age. Our result is in agreement with that reported by Qasemi et al. (2021), [37] who found that the clearance of cfDNA is less efficient in women with endocrine pathology, such as PCOS. FSH and AMH are a couple of hormones that interact to produce follicles and oocytes in the ovary [38]. In the FF of the PCOS and non-PCOS women in our study, we found a negative association between cfDNA and both FSH and AMH levels.

### 3.1. Assessment by Female Age

The relative ratio of cf-mtDNA to cf-nDNA was assessed in relation to female age. More specifically, a comparison of 57 FF cfDNA obtained from a group of women between 23–35 years (average age 31 years,) and 44 FF cfDNA from an older group with a range between 36–43 years (average age 38.8 years,) was analyzed with real-time PCR. The data analysis resulted in a statistically significant increase (*p* = 0.000009) in the amount of the ratio mtDNA/nDNA in FF samples from the older women (Figure 1). The relative amounts of mtDNA/nDNA ratio for the female age groups under investigation are summarized in Table 4 and illustrated in Figure 1.

### 3.2. Assessments Based on Polycystic Ovary Syndrome and the Age of the Female

A significant difference (*p* = 0.0000003) in the levels of mtDNA to nDNA ratio was also observed in the three categories of women compared. The first two groups included the reproductively younger (*n* = 34) and older (*n* = 36) women, respectively, and the third group included women with PCOS (*n* = 31). Once again, data analysis clearly showed a statistically significant increase in the ratio of FF samples from the reproductively older women. As concerns women with PCOS, no statistical difference was observed. It appeared that all samples followed the age pattern illustrated in Figure 1. For the female groups included in the study, the relative mtDNA/nDNA ratio amounts are listed in Table 5 and presented in Figure 2 and Figure 3.

### 3.3. Comparison between Pregnant and Not Pregnant Women

A similar comparison was made between the FF cf-mtDNA/cf-nDNA ratio and the cf-mtDNA relative content (RCN), of women with or without successful pregnancy. However, the analysis demonstrated a nonstatistically significant difference (*p* = 0.7) (*p* = 0.3) (Figure 4 and Figure 5).

### 3.4. Correlation of the mtDNA Relative Copy Number (RCN) in Individual Groups

An increased risk of age-related diseases such as cancer, cardiovascular disease, and neurological disorders is linked to the reduction in the mtDNA copy number with age. This is due to the vital role of mitochondria in cellular functions, such as energy production, and others that are necessary for preserving good cellular function and tissue homeostasis. Overall, the loss of mtDNA copies with age can influence both a woman’s ability to get pregnant and the children’s health. We, therefore, compared the relative mtDNA content in these samples. We observed that this number is reduced in reproductively older women, a finding consistent with the currently available research. The relative mtDNA copy number was calculated based on the ΔΔCt method, as described in materials and methods. We evaluated the samples to assess the multiplicity of times they are expressed as compared to the control group, which consisted of reproductively younger women who had successful pregnancies, using the 2^−ΔΔCt^ formula. A value of 2^−ΔΔCt^ > 1 indicates that the group under examination exhibits higher levels of mtDNA than the control group. It has been previously reported that FF cf-mtDNA levels are higher in PCOS than in women without this condition. According to Qasemi et al. (2021), [37] in women without a specific reproductive problem, FF cf- mtDNA can be used as a biomarker for potential ART outcomes. As shown in Table 6 and Figure 6, our data are in agreement with these authors, suggesting that abnormal mtDNA accumulation in this pathological situation may cause increased FF cf-mtDNA. The data are presented in Table 6 and illustrated in Figure 6.

Interestingly, we observed a significant decrease (*p* < 0.0001) in the relative mtDNA content with advancing age (Figure 7) and a positive correlation (*p* < 0.0001) with the number of mature oocytes (MII stage), indicating that this parameter can be a useful biomarker of oocyte quality and ovarian ageing (Figure 8).

## 4. Discussion

Mitochondria are crucial for numerous biological processes and their involvement in human reproduction is particularly significant. Mitochondria are essential for all cellular functions, including the generation of ATP, embryonic development, and oocyte fertilization [39]. The distinctive feature of mitochondria is that they are vertically maternally transmitted. The oocyte has more mitochondria because the egg cell supplies the developing embryo with far more cytoplasm than the sperm. The sperm contributes half of the genetic material to the embryo but, after fertilization, its mitochondria are frequently destroyed or damaged. Studies demonstrating that errors in mitochondrial activity can result in a variety of reproductive pathologies, including infertility, miscarriage, and developmental abnormalities, have raised awareness of the significance of mitochondrial function in human reproduction [40].

Although mitochondria have their DNA, nuclear DNA also plays a significant role in most of the protein production and maintenance in mitochondria [41,42]. The relative abundance of mitochondrial DNA compared to nuclear DNA within a cell is referred to as the mtDNA/nDNA ratio. Changes in the mtDNA/nDNA ratio and mitochondrial number in ageing females are influenced by several factors. One significant factor is the accumulation of mtDNA mutations over time. Mitochondrial DNA is more vulnerable to oxidative damage, and its DNA repair mechanisms are less efficient [43]. As a result, mtDNA accumulated mutations may become more prevalent with age. These mutations may interfere with the mitochondria’s ability to replicate and maintain, which could be detrimental to fertility [44]. Age-related declines in mitochondrial content can also be caused by several factors, such as hormonal changes reduction in mitochondrial biogenesis, lifestyle, genetics, and environmental influences [45].

Apoptosis and mitophagy are two cellular processes that are closely related, with a complicated and unclear interaction. Nevertheless, it has been put forward that mitophagy may function as a prosurvival process by eliminating defective mitochondria that could trigger apoptosis [46]. Furthermore, it has been demonstrated that inhibiting mitophagy can promote apoptosis [47]. Both the effectiveness of apoptosis and mitophagy can decrease as women age. An accumulation of damaged mitochondria in the cells could result from this decline. Age-related increases in damaged or malfunctioning mitochondria lead to a higher mtDNA/nDNA ratio. Both ovarian follicular growth and atresia (follicle degeneration) depend heavily on apoptosis. Only a percentage of the follicles that form during folliculogenesis mature and release a viable oocyte. The remaining follicles deteriorate and undergo apoptosis [48]. According to investigations on the cf-DNA’s integrity in follicular fluid, the release of cf-nDNA into the follicular fluid is one of the several cellular and molecular pathways that are activated during apoptosis and up to 85% of the cfDNA originate from cell apoptosis [24].

Based on the findings of our study, there is an apparent contradiction about the fact that the follicular mtDNA/nDNA ratio increases with age, although the number of mitochondria itself has indeed decreased. It is already known that the amount of cfDNA in the follicular fluid may be a good indicator of the oocyte quality inside the follicle and the level of follicular apoptosis. High amounts of cfDNA have been linked to decreased oocyte developmental capacity and higher follicular atresia [9].

To enhance the effectiveness of infertility therapies, new techniques for assessing oocyte competency and embryo viability should be developed. Studies focusing on the mitochondrial function, granulosa and cumulus cells, oocytes, and embryos, as well as follicular fluid, will likely lead to novel findings. Most of the research has established a link between the success of ART and the capacity of oocytes for supporting preimplantation development and pregnancy. The identification of a high-quality embryo using trustworthy molecular technology and the measurement of the success of ART are the two main goals of the biomarkers employed in human reproductive medicine. However, to make these techniques as efficient as possible, it is crucial to search for the factors that affect the quality of the follicles and, subsequently, of the oocytes. There have been numerous evaluations of mtDNA in IVF clinical outcomes assessment [49]. Higher trophectoderm quality, effective implantation outcomes, and successful embryonic development into blastocysts are also related to an increased cf-mtDNA/cf-nDNA ratio [50]. cfDNA as a biomarker has created new opportunities for noninvasive diagnostic tool applications in biological processes [11]. Both the cell-free nuclear DNA (cf-nDNA) and the cell-free mitochondrial DNA (cf-mtDNA) are considered to be quite valuable tools in the diagnosis and prognosis of cancer, as well as in prenatal diagnosis [12,13]. Prenatal testing is one of the most well-known uses of cfDNA since it can be used to find genetic anomalies in a growing fetus.

The current study aims to highlight mitochondria as a potential high-sensitivity prognostic tool for assessing an embryo’s viability in artificial fertilization and to ascertain whether their use can be implemented in ART. We intended to provide an alternative, non-time-consuming, non-cost-intensive, and noninvasive approach to measuring the amount of cell-free mitochondrial DNA (cf-mtDNA) by calculating the ratio cf-mtDNA/cf-nDNA in FF samples of women undergoing follicular aspiration for IVF/ICSI. We found that the FF cf-mtDNA/cf-nDNA ratio level was higher in women, who also had the highest rates of successful IVF. The relative ratio of cf-mtDNA to cf-nDNA content in human follicular fluid was positively correlated with age and the implantation potential only for females without a specific reproductive pathology but not among those with PCOS, demonstrating that, based on our preliminary findings, the relative ratio of cf-mtDNA to cf-nDNA content in human FF may serve as a potential predictor for assessing the age-related yield of the corresponding oocyte in IVF. However, further studies with larger sample sizes are needed to confirm these findings.

Last, but not least, it should be highlighted that the majority of experts in the field of medicine agree that mitochondria are an excellent predictor. Despite the significance of mitochondria in human reproduction, their function and regulation remain poorly understood. Many studies have demonstrated the potential clinical utility of cfDNA (cf-mtDNA and cf-nDNA) analysis in follicular fluid in ART for the quality of the oocyte and the possibility of a successful pregnancy; however, more studies are required. Fertility, time to pregnancy, and ageing of the oocytes appear to be influenced by cf-mtDNA copy counts. Even if the cf-mtDNA copy numbers can be employed as a clinical tool for diagnosing and treating reproductive pathologies, further study is required to fully understand the mechanisms underlying these correlations. Finally, due to mtDNA’s significant degree of variability, the implementation of other markers, such as the cf-mtDNA and cf-nDNA ratio, hormones, and morphology, can be exploited in conjunction with other biomarkers.

## 5. Conclusions

Biomarkers are enhancing what is already known in prognosis, diagnosis, and treatment. They are useful tools with important biological and therapeutic applications in many medical disciplines. Preserving the follicular fluid milieu is crucial for producing suitable eggs and, hence, well-developed embryos. Trustworthy relevant biomarkers and noninvasive diagnostic tests are essential in the field of human reproductive medicine. In recent years, there has been growing interest in the role of cfDNA in human reproduction. Due to genomic DNA degradation, low concentration, and high fragmentation, studying cfDNA is typically challenging but it might be enhanced to serve as a useful tool in medically assisted reproduction. The follicular fluid, as the immediate microenvironment of the egg, provides all the potentially necessary ingredients to nourish the oocytes before fertilization. However, until recently, the selection of the embryos with the highest implantation potential during assisted reproduction procedures was taking into account the morphological criteria of the eggs. Since follicular fluid is only obtained when oocytes are aspirated from follicles, it is commonly acknowledged that this procedure is not entirely invasive. As a result, the qualitative and quantitative determination of its cf-nDNA and cf-mtDNA is rapid, exhibits great sensitivity, and is simple to carry out. Furthermore, it provides a general overview of the quality of the follicular microenvironment, enhancing the success of ART. It is generally known that the mother’s age has an adverse relationship with the likelihood that an egg will develop into a viable embryo, a fact that is strongly confirmed by the significant difference in the success rate of IVF treatment for reproductively older women compared to younger. However, the decrease in cf-mtDNA with age seen in FF samples during the current study raises the question of whether mitochondria might play a direct role in female fertility and age. Last, but not least, to sum up, more investigation is needed to thoroughly examine the potential applications of cell-free DNA, nDNA, mtDNA, and the mt/n ratio, in reproductive biology and to develop more effective diagnostic and therapeutic approaches based on this biomarker.

## Figures and Tables

**Figure 1 genes-14-01504-f001:**
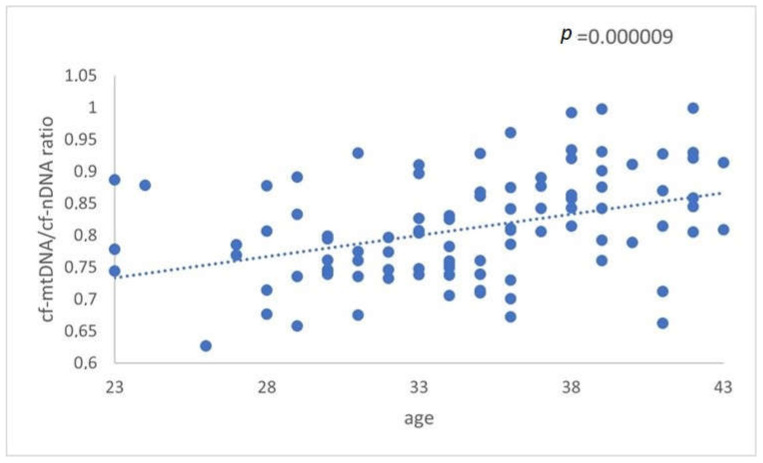
The relationship between cf-mtDNA/cf-nDNA quantity and female age. Data obtained during quantitative real-time PCR analysis of FF samples taken from women undergoing IVF demonstrated a statistically significant increase (*p* = 0.000009) in the level of ratio about advancing female age.

**Figure 2 genes-14-01504-f002:**
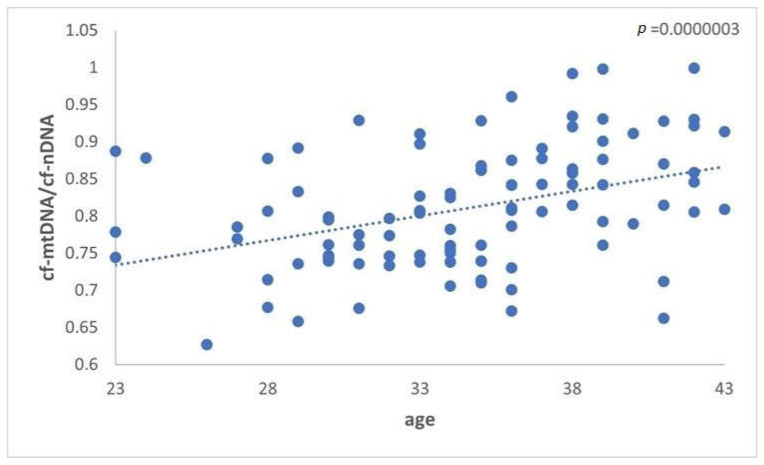
The relationship between cf-mtDNA/cf-nDNA quantity in the 3 groups. Data obtained during quantitative real-time PCR analysis of FF samples taken from women undergoing IVF demonstrated a statistically significant increase (*p* = 0.0000003) in the level of ratio compared to advancing female age. Specifically, the *p*-value was further decreased in older women. No apparent difference was indicated for women with PCOS.

**Figure 3 genes-14-01504-f003:**
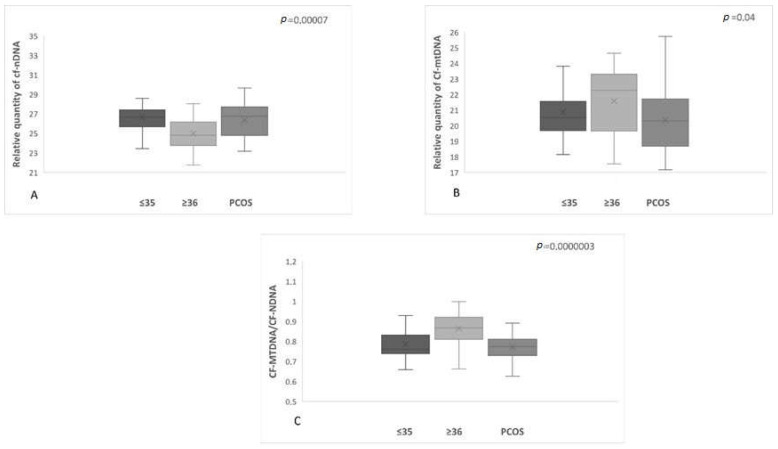
(**A**). The relationship between cf-nDNA quantity, female age, and PCOS demonstrated a statistically significant increase (*p* = 0.00007) in women with PCOS. (**B**). The relationship between cf-mtDNA quantity, female age, and PCOS demonstrated a statistically significant increase (*p* = 0.04) in women ≥36 years of age. (**C**). The relationship between the ratio cf-mtDNA/cf-nDNA, female age, and PCOS demonstrated a statistically significant increase (*p* = 0.0000003) in the reproductively older women.

**Figure 4 genes-14-01504-f004:**
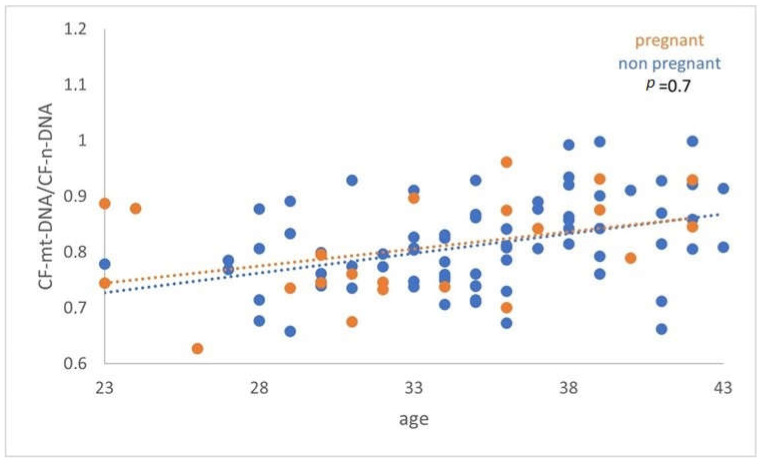
The analysis did not demonstrate a statistically significant difference (*p* = 0.7) between women with positive and negative pregnancy tests.

**Figure 5 genes-14-01504-f005:**
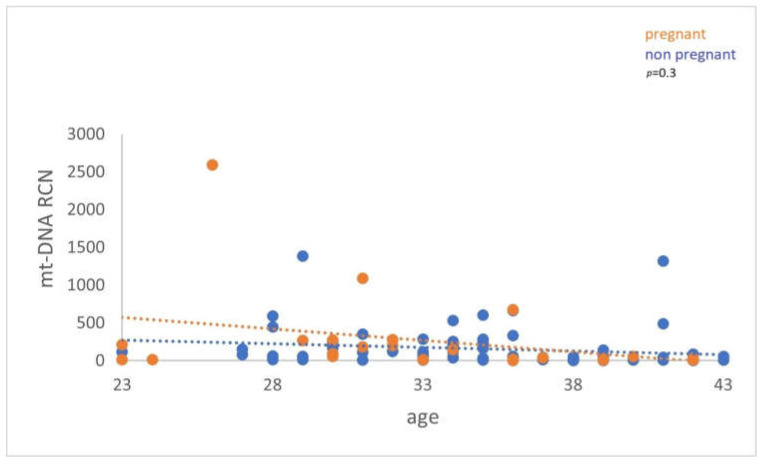
cf-mtDNA relative copy number (RCN) between women with positive and negative pregnancy tests. Data analysis has not demonstrated a statistically significant difference (*p* = 0.3).

**Figure 6 genes-14-01504-f006:**
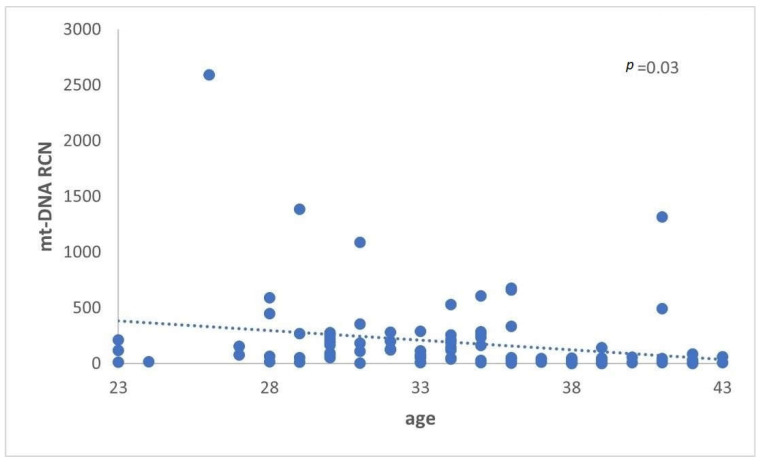
Cf-mtDNA relative copy number (RCN) concerning female age. Data analysis has demonstrated a statistically significant (*p* = 0.03) decrease in the cf-mtDNA copy number with advanced female age.

**Figure 7 genes-14-01504-f007:**
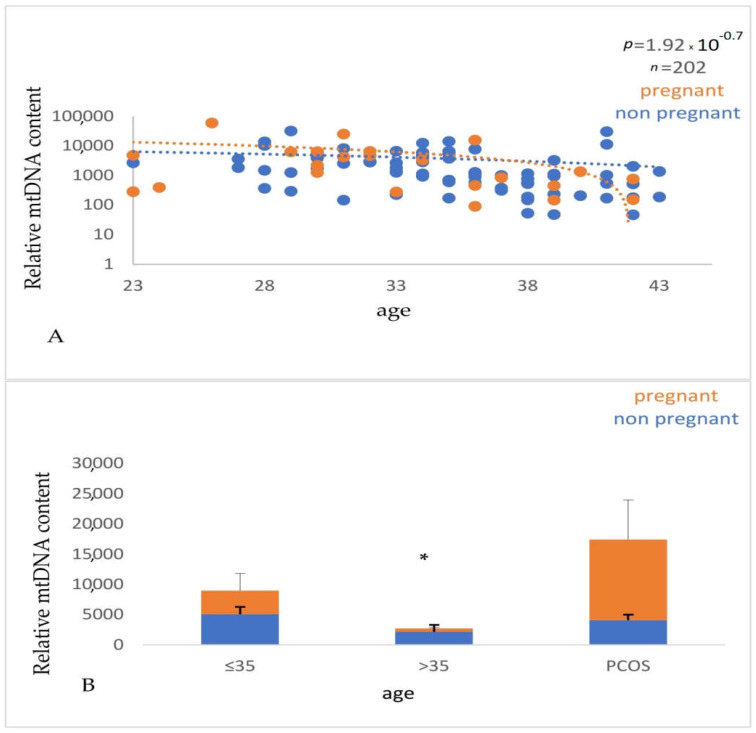
(**A**). Correlation between the relative mtDNA content and the age of the women. Values corresponding to women with positive and negative pregnancy tests are shown with orange and blue circles, respectively. The data were obtained from qPCR analysis of cell-free DNA isolated from the FF samples that were taken from women undergoing IVF and analyzed by the 2^−ΔCt^ method. A significant decrease (*p* < 0.0001) in the relative mtDNA content with advancing age was observed. (**B**). Graphical representation of the relative mtDNA content with the age of the women. The orange part of the bars corresponds to pregnant women and the blue part to nonpregnant women. * A significant difference (*p* < 0.05) of the relative mtDNA content between the young and older women was observed.

**Figure 8 genes-14-01504-f008:**
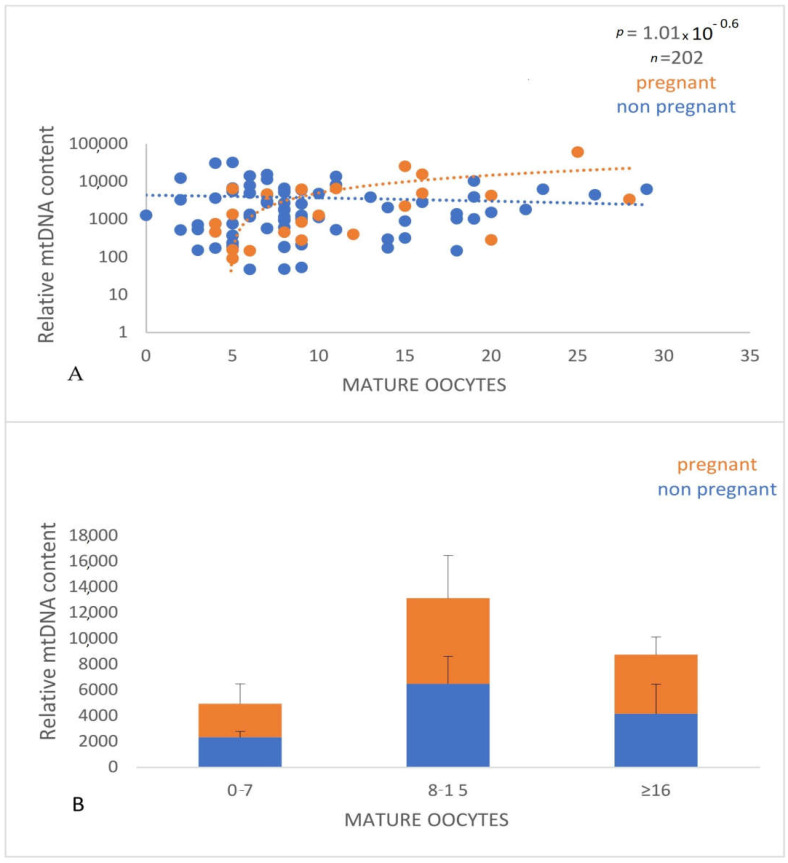
(**A**). The relationship between the relative mtDNA content and the number of mature (MII) oocytes. Values corresponding to women with positive and negative pregnancy tests are shown with orange and blue circles, respectively. The data were derived as described in Figure 7. (**B**). Graphical representation of the relative mtDNA content with the number of mature (MII) oocytes. The orange parts of the bars correspond to pregnant women and the blue parts to nonpregnant women. A significant difference (*p* < 0.05) of the relative mtDNA content between women who had a relatively small (0–7 and 8–15) number of mature oocytes was revealed.

**Table 1 genes-14-01504-t001:** General characteristics of the women who participated in the study, including the number of oocytes and granulosa cells. SD, standard deviation.

Group	No of Samples	Mean Age ± SD	Mean No of Oocytes ± SD	No of Granulosa Cells
≤35	34	31.4 ± 3.5	10.7 ± 4.1	112.213
≥36	36	39.3 ± 2.1	10.1 ± 5.7	135.000
PCOS	31	32.0 ± 3.6	18.0 ± 8.1	199.298

**Table 2 genes-14-01504-t002:** Hormone concentrations in each group of participants. Clinical characteristics of the women who participated in the study. AMH, FSH, and LH blood-serum levels were determined in a cycle preceding IVF treatment. The estradiol blood-serum levels were measured 2 days before oocyte retrieval. SD, standard deviation; AMH, anti-Mullerian hormone; FSH, follicle-stimulating hormone; LH, luteinizing hormone; Estr, estrogen.

Group	AMH (ng/mL) ± SD	FSH ± SD	LH ± SD	Estr (pg/mL)
≤35	3.58 ± 4.2	6.11 ± 1.5	7.98 ± 7.7	2398.73
≥36	4.56 ± 6.9	7.16 ± 2.7	6.45 ± 11.6	2226.96
PCOS	10.58 ± 18.3	6.45 ± 1.1	14.65 ± 14.6	2901.20

**Table 3 genes-14-01504-t003:** Oligonucleotide sequences for cf-Mt/cf-N determination used for the real-time quantitative PCR, the corresponding Tm, and the size of the target area.

Primer Name	Sequence (Forward/Reverse)	Amplicon bp	Tm
Nuclear gene (RPE)	5′-ATAGGAAGCCAGAGAAGAGAGACT-3′ 5′-TCTATCTCTGCGGACTTTGAGCAT-3′	200 bp	60 °C
Mitochondrial gene	5′-TAGAGGAGCCTGTTCTGTAATCG-3′ 5′-TAAGGGCTATCGTAGTTTTCTGG-3′,	205 bp	59 °C

**Table 4 genes-14-01504-t004:** The table displays the mean values for the age and the ratio of the samples.

Group	No of Samples	Mean Age ± SD	Mean Cf-mtDNA/Cf-nDNA ± SD
Reproductively younger women	57	31.0 ± 3.3	0.77 ± 0.06
Reproductively older women	44	38.8 ± 2.3	0.85 ± 0.08

**Table 5 genes-14-01504-t005:** Data are represented as mean ± SD.

Group	No of Samples	Age ± SD	Cf-mtDNA/ Cf-nDNA ± SD
Reproductively younger women	34	31.4 ± 3.5	0.78 ± 0.06
Reproductively older women	36	39.3 ± 2.1	0.86 ± 0.07
PCOS women	31	32.0 ± 3.6	0.77 ± 0.06

**Table 6 genes-14-01504-t006:** The quantitative real-time PCR analysis resulted in the below outcomes. There are also highlights of the ratio cf-mtDNA/cf-nDNA and the relative amount of mitochondria. Data are represented as mean ± SD.

Group	cfDNA (ng/μL) ± SD	Cf-nDNA (Ct Average) ± SD	Cf-mtDNA (Ct Average) ± SD	Cf-mtDNA/ Cf-nDNA ± SD	2 × 2^ΔCt^ (RCN)	2^−ΔΔCt^
≤35	10.38 ± 9.1	26.62 ± 1.2	20.88 ± 1.5	0.78 ± 0.06	207	2.28
≥36	7.68 ± 7.2	24.99 ± 1.7	21.59 ± 2.1	0.86 ± 0.07	76	9.26
PCOS	8.67 ± 5.9	26.37 ± 1.9	20.36 ± 2.3	0.77 ± 0.06	289	1.42
Control group	12.8 ± 6.2	26.3 ± 0.9	20.7 ± 1.4	0.79 ± 0.07	166	

## Data Availability

Data is unavailable due to privacy or ethical restrictions.

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
