# Peer review of "The Ratio of cf-mtDNA vs. cf-nDNA in the Follicular Fluid of Women Undergoing IVF Is Positively Correlated with Age"

_genes, 2023, doi:10.3390/genes14071504_

Round 1
Reviewer 1 Report
In this report, the authors presented a study of the cf-mtDNA / cf-nDNA ratio, in follicular fluid (FF) of women undergoing IVF, to generate a molecular fingerprint of oocyte quality. The values of this ratio were measured and compared in three groups of women: women with polycystic ovary syndrome (PCOS), women younger than 35, and women older than 36. The authors used the correct method. They performed qPCR to quantify the ratio using nuclear- and mitochondrial-specific primers and analyzed for potential correlation with age and pregnancy rates. The analysis showed that FF-cf-mtDNA levels were lower in the advanced age group than in the PCOS and non-PCOS groups. They observed a positive correlation between FF-cf-mtDNA and the number of mature oocytes.
The authors' data show that the relative ratio of cf-mtDNA to cf-nDNA content in human FF can be an effective predictor for assessing the age-related yield of the corresponding oocyte in IVF. This is too bold a statement, since only 101 women were studied.
Please correct line 23-24 - the data on women's ages are not consistent with Table 1.
Author Response
Dear Reviewer,
We appreciate your insightful comments and suggestions, which have helped us to improve the quality of our manuscript. Please find below our responses to your comments:
- Comment on the boldness of the statement: We agree with your observation that our statement regarding the predictive value of the cf-mtDNA to cf-nDNA ratio in human FF was too assertive given the sample size of our study. We have revised this statement in our manuscript to reflect the preliminary nature of our findings. The revised statement now reads: "Our preliminary findings suggest that the relative ratio of cf-mtDNA to cf-nDNA content in human FF may serve as a potential predictor for assessing the age-related yield of the corresponding oocyte in IVF. However, further studies with larger sample sizes are needed to confirm these findings."
- Comment on the inconsistency between the text and Table 1: We appreciate your attention to detail in identifying this inconsistency. Upon review, we have corrected the data on women's ages in lines 23-24 to match the data presented in Table 1. We apologize for any confusion this may have caused and appreciate your understanding.
We hope that these revisions adequately address your concerns. We are open to further feedback and discussion to improve our manuscript.
Thank you for your time and consideration.
Best regards,
Georgia Tsirka

Reviewer 2 Report
Interesting observational study on follicular fluid biomarkers.
While this does not add to clinical practice, it is well presented and a good read.
The emphasis on more research required on follicular fluid markers is well highlighted
Author Response
Dear Reviewer,
We appreciate your positive feedback on our study and are glad to hear that you found it interesting and well-presented. We agree with your observation about the need for more research on follicular fluid markers, and we hope that our study contributes to this growing field of research.
We are grateful for your time and consideration in reviewing our manuscript. Your comments are encouraging, and we look forward to contributing further to this area of study.
Best regards,
Georgia Tsirka
